# Analysis of Antimicrobial Resistance in Bacterial Pathogens Recovered from Food and Human Sources: Insights from 639,087 Bacterial Whole-Genome Sequences in the NCBI Pathogen Detection Database

**DOI:** 10.3390/microorganisms12040709

**Published:** 2024-03-30

**Authors:** Ashley L. Cooper, Alex Wong, Sandeep Tamber, Burton W. Blais, Catherine D. Carrillo

**Affiliations:** 1Research and Development, Ottawa Laboratory (Carling), Canadian Food Inspection Agency, Ottawa, ON K1A 0C6, Canada; burton.blais@inspection.gc.ca; 2Department of Biology, Carleton University, Ottawa, ON K1S 5B6, Canada; alexwong@cunet.carleton.ca; 3Microbiology Research Division, Bureau of Microbial Hazards, Health Canada, Ottawa, ON K1A0K9, Canada; sandeep.tamber@hc-sc.gc.ca

**Keywords:** antimicrobial resistance (AMR), foodborne bacteria, food production, food pathogen surveillance, ESKAPEE pathogens, biocide resistance, metal resistance

## Abstract

Understanding the role of foods in the emergence and spread of antimicrobial resistance necessitates the initial documentation of antibiotic resistance genes within bacterial species found in foods. Here, the NCBI Pathogen Detection database was used to query antimicrobial resistance gene prevalence in foodborne and human clinical bacterial isolates. Of the 1,843,630 sequence entries, 639,087 (34.7%) were assigned to foodborne or human clinical sources with 147,788 (23.14%) from food and 427,614 (76.88%) from humans. The majority of foodborne isolates were either *Salmonella* (47.88%), *Campylobacter* (23.03%), *Escherichia* (11.79%), or *Listeria* (11.3%), and the remaining 6% belonged to 20 other genera. Most foodborne isolates were from meat/poultry (95,251 or 64.45%), followed by multi-product mixed food sources (29,892 or 20.23%) and fish/seafood (6503 or 4.4%); however, the most prominent isolation source varied depending on the genus/species. Resistance gene carriage also varied depending on isolation source and genus/species. Of note, *Klebsiella pneumoniae* and *Enterobacter* spp. carried larger proportions of the quinolone resistance gene *qnrS* and some clinically relevant beta-lactam resistance genes in comparison to *Salmonella* and *Escherichia coli*. The prevalence of *mec* in *S. aureus* did not significantly differ between meat/poultry and multi-product sources relative to clinical sources, whereas this resistance was rare in isolates from dairy sources. The proportion of biocide resistance in *Bacillus* and *Escherichia* was significantly higher in clinical isolates compared to many foodborne sources but significantly lower in clinical *Listeria* compared to foodborne *Listeria*. This work exposes the gaps in current publicly available sequence data repositories, which are largely composed of clinical isolates and are biased towards specific highly abundant pathogenic species. We also highlight the importance of requiring and curating metadata on sequence submission to not only ensure correct information and data interpretation but also foster efficient analysis, sharing, and collaboration. To effectively monitor resistance carriage in food production, additional work on sequencing and characterizing AMR carriage in common commensal foodborne bacteria is critical.

## 1. Introduction

Antimicrobials, including antibiotics, biocides, and metals, are arguably one of the most important discoveries in the history of medicine. The introduction of antibiotics (such as penicillin) resulted in a shift in the leading causes of death from infectious diseases, including gastroenteritis, pneumonia, and tuberculosis, to non-communicable diseases, such as heart disease, cancer, and stroke [1].

In addition to applications in human medicine, antimicrobials are used to treat disease in agriculture and food animal production [2,3,4]. Antimicrobial use in agriculture is necessary for plant, animal, and human health, as large-scale agri-food production practices often involve high population densities. Metal compounds containing copper, zinc, cadmium, and arsenic are used in agriculture; meanwhile, clinical applications include mercury, nickel, copper, aluminium, titanium, and zinc-based metal-containing products [5,6,7,8,9,10]. In addition to antibiotics and metals, biocides (disinfectants or sanitizers) are often utilized during food production. Generally, biocides are defined as substances that are formulated to be harmful to living organisms [11]. Biocides are used to clean and disinfect equipment and surfaces in health care, farming, and food production settings; as decontaminants on carcass surfaces; and as preservatives in cosmetics, pharmaceuticals, and foods in order to control pathogenic and spoilage microorganisms [11,12].

Unfortunately, bacteria have evolved various strategies, including intrinsic and acquired mechanisms, to avoid antimicrobials. Consequently, antimicrobial resistance (AMR) is commonly observed in microorganisms. The anthropogenic use of antimicrobials is believed to be a contributing factor in the evolution and transmission of AMR by creating selective pressures for persistence [13]. Food crops and animals harbour bacteria that are pathogenic to humans [14], and the spread of bacteria from these sources to food products is extensively documented [14,15,16,17]. The pathogenic and commensal bacteria of food microbiota(s) can inhabit and spread between multiple environments, including agricultural, food processing, aquatic, and clinical settings, where they could potentially acquire and transmit virulence and AMR genes (ARGs) (Figure 1). Among the AMR bacteria, the ESKAPEE pathogens—an acronym for *Enterococcus faecium*, *Staphylococcus aureus*, *Klebsiella pneumoniae*, *Acinetobacter baumannii*, *Pseudomonas aeruginosa*, *Enterobacter* species, and *Escherichia coli*—are of particular concern due to their increasing resistance to antibiotics used in human medicine [18,19,20]. These bacteria not only cause serious healthcare-associated infections but have also been detected in food-producing animals and related environments, highlighting the potential for foodborne transmission to humans [21,22].

The term “antibiotic” refers to substances produced by microorganisms but does not typically encompass synthetic antimicrobials (such as sulphonamides and quinolones) or medicines used to prevent and treat bacterial infections. As the term “antimicrobial” can refer to all agents that act against microbial organisms, metals and biocides are technically also antimicrobials. As such, for this publication, the term antibiotic will refer to all chemotherapeutic antibiotics used to treat infection, including all antimicrobials that are not metals or biocides. Biocides will refer to disinfectants and sanitizer products with more varied applications, such as quaternary ammonium compounds, chlorine-releasing agents, and peroxygens, which are not selective enough to be used within body tissues, but will not include antibiotics (many of which are technically harmful to living microorganisms) [23].

Antibiotics are used in food animal production to increase feed efficacy, as growth promoters, and prophylactically to prevent disease circulation, and evidence suggests that this use in animals has contributed to the development and spread of AMR in humans [2,3,4,13,24]. As with antibiotics, increased metal resistance has been observed in bacteria isolated from animals whose feed has been supplemented with metal compounds [25]. In addition, many metalloids including mercury, copper, and zinc have been released into the environment through anthropogenic activities [10,26,27]. Similarly, increases in biocide resistance have been observed [11,28,29]. Genes encoding resistance to biocides, including quaternary ammonium compounds (QACs), have been found in Gram-negative and Gram-positive bacteria [30,31,32,33,34]. The spread of resistance to biocides used in food production has been observed [35]. As biocide and metal resistance may develop through increased efflux, or the acquisition of mobile genetic elements (MGEs) encoding resistance genes, there is concern that the development of bacterial biocide or metal resistance may also result in increased bacterial antibiotic resistance.

In fact, the co-selection of biocide, metal, and antimicrobial resistance has been observed among pathogens and other bacteria [10,11,36]. The use of biocides and preservatives may increase ARG transfer among bacteria as well as co-select for multi-drug-resistant (MDR) strains [11,35,37,38,39]. Studies have reported an association between biocide use in poultry and egg production and the isolation of biocide-tolerant and antimicrobial-resistant *Salmonella* spp. [40,41,42]. Nonetheless, as with antibiotic resistance, some studies suggest that repeated disinfectant use in food processing and agricultural environments does not select for biocide or antimicrobial resistance [43,44]; furthermore, a recent study found that the natural evolution of ARGs led to the maintenance of bacterial resistance, despite the reduction in antimicrobial use [45].

The National Centre for Biotechnology Information (NCBI) Pathogen Detection database (NPDD) resource “integrates bacterial and fungal pathogen genomic sequences from numerous ongoing surveillance and research efforts” and includes data from food, environmental, and clinical sources of both human and animal origin [46]. Previous studies have utilized the NPDD whole-genome sequence (WGS) collection and other public sequence repositories to investigate transmission sources and genotypes associated with food contamination and foodborne illness in *Salmonella* and *Listeria*, the resistome and virulence analysis of *Campylobacter* spp., specific resistance genes in bacteria from meat products in six US states, and the multivariate analysis of ARGs in eight different countries [47,48,49,50,51]. However, to the best of our knowledge, there are currently no studies that utilize this resource to explore the prevalence of AMR across different bacterial genera originating from various food categories.

The objective of this study is to better understand and explore the strengths and limitations of available bacterial genomic data from food sources. Additionally, we aim to identify any existing gaps and expand our current knowledge on AMR data pertaining to foodborne bacteria. Through the analysis of metadata from 639,087 bacterial genomes from the NPDD, our study seeks to offer a comprehensive examination of the distribution of ARGs, including metal and biocide resistance, in foodborne bacteria as compared to clinical isolates. This analysis includes bacterial isolates from diverse countries and food categories, offering a broad overview of the abundance of various antimicrobial classes and a detailed examination of some clinically significant ARG families. By comparing ARGs in food isolates with those found in clinical isolates, this research aims to uncover insights into the prevalence and distribution of priority AMR in bacteria recovered from foods.

## 2. Materials and Methods

### 2.1. Retrieval of Bacterial Sequence Metadata from NCBI/NPDD Analysis Pipeline

Data were obtained from the NPDD on 17 November 2023 [46]. Bacterial genomic sequence analysis results in the form of AMR metadata files were downloaded from the NCBI Pathogen Detection FTP for select organisms that have been isolated from food products (Appendix A). As information regarding isolate identifiers, isolation source lot numbers, patient identifiers, etc., was not available for all sequences, it is likely that some sequences included in this analysis are duplicates or clonal in origin. The final number of genomes analysed from each source for each genus/species are listed in Table 1. Metadata table versions, the total number of sequences, and download date information are available in Appendix A.

### 2.2. Isolation Source Categorization

Isolation sources for the NCBI Pathogen Detection metadata table “epi_type” environmental/other category were manually curated for each organism based on the provided sequence submission information under “isolation_source”. Information regarding the assignment and definitions of source categories is summarized in Table 2, and available at https://github.com/OLC-Bioinformatics/source_and_resistance_categorizer.git, (last updated November 2023). under the “Source Definitions” section [52]. The National Institute of Health (NIH) and NCBI currently provide interagency food safety analytics collaboration (IFSAC) CDC categorization in the metadata files in the pathogen detection database [53,54]. Where IFSAC categories were not provided, all unique values from the “isolation_source” column of downloaded metadata tables were extracted and assigned to source categories (e.g., chicken breast was assigned to meat/poultry, cheese was labelled as dairy, lettuce was labelled as fruit/vegetables, etc.). These source categories were used to append simplified, curated, source information to the NCBI Pathogen Detection metadata tables using custom Python scripts (available at [52]). Where categorization existed for both IFSAC and the manually curated isolation source data, the IFSAC category was selected by the script for the final ‘Source’ column. Only a subset of source assignments related to food products were investigated and included in this study (Table 2). Clinical data were defined as data entries with the “epi_type” designated as clinical and “host” designated as Homo sapiens. A more comprehensive list of source category information and a dictionary file containing all curated sources from metadata are available in the previously mentioned github repository.

### 2.3. Antimicrobial Resistance Categorization

Antimicrobial resistance class/type was simultaneously assigned using the custom Python script mentioned above. Briefly, the AMRFinderPlus database Reference Gene Catalog (version 3.11) was downloaded from the NCBI FTP (https://ftp.ncbi.nlm.nih.gov/pathogen/Antimicrobial_resistance/AMRFinderPlus/database/latest/ accessed on 17 November 2023) and used to separate resistance genes into antibiotic, biocide, and metal resistance categories [55]. Genes belonging to the antibiotic category were further divided based on resistance to specific antibiotic classes (e.g., aminoglycoside, ß-lactam, tetracycline, etc.). These gene class assignment lists were separated and included in the resistance_genes.csv dictionary file used with the custom Python script mentioned above to append resistance class information to NPDD metadata tables based on genes listed in the ‘AMR_genotypes’ and ‘stress_genotypes’ columns.

### 2.4. Enumeration of Resistance by Isolation Source

Following isolation source and resistance class assignment for each of the genera and species listed above, the number of isolate sequences for each genus/species from each source encoding each resistance class were tallied. For select resistance classes, the numbers of each genus/species encoding specific resistance gene alleles of interest were also determined. The majority of the gene families or gene alleles counted were for known transferrable ARGs that confer clinically important resistance, with the exception of the quinolone class, where some genera in the NCBI Pathogen Detection database included data for chromosomal point mutations conferring resistance (e.g., *gyrA*, *parC*, and *parE* mutations conferring quinolone resistance). For vancomycin resistance, sequences were tallied as positive (vanA, vanB, vanG, vanR-A/vanS-Pt) if they encoded all genes in the operon required for that cluster [56].

### 2.5. Statistical Analysis

For each genus/species, the comparison of resistance proportions between different isolation sources was conducted using the Fisher’s exact test with the Benjamini and Hochberg (BH) adjustment in R version 4.3.0 [57] and the rstatix package for pairwise comparisons [58]. The Fisher’s exact test with BH correction was also used to compare proportions of isolates encoding each antimicrobial class with the proportion encoding both the antimicrobial class and biocide resistance.

For each genus, the association of resistance class with isolation source was conducted using a Chi square test. Data were subset by genus and isolation source for all sources with at least one isolate/sequence. The Chi square test was then performed on contingency tables of resistance class versus source using the chisq.test function from the core R Stats package version 4.3.0. To evaluate the association of resistance classes with isolation source, Pearson standardized residuals from Chi square tests were plotted using the corrplot package with the “is.corr” flag set to FALSE in R version 4.3.0 [57,59].

## 3. Results

A total of 639,087 isolate genome sequences from human clinical (*n* = 491,299, 76.88%) and food (*n* = 147,788, 23.12%) sources were selected from the NCBI Pathogens dataset (total = 1,843,630 genomes) based on the completeness of the isolation source information provided (Table 1 and Appendix A). Of these, the majority of the genomes from all sources were *Salmonella* (28.94%), *Escherichia* (16.94%), *Staphylococcus* (11.72%), *Klebsiella* (9.60%), *Campylobacter* (7.92%), and *Listeria* (4.21%) species. The other 19 genera each accounted for less than 3.5% of the sequences analysed, for a combined total of 20.67% (Table 1). Most of the genomes from food sources were *Salmonella* (47.88%), *Campylobacter* (23.03%), *Escherichia* (11.79%), and *Listeria* (11.3%), with the remaining 20 genera only accounting for a combined 6% of food isolates. Most of the foodborne isolate sequences were from meat/poultry (64.45%) and multi-ingredient food (20.23%) sources, although this varied by genus and species (Table 1). The next highest isolation sources were fish/seafood (4.40%), dairy (3.76%), and fruit/vegetables (4.64%) which were not well represented in comparison to meat and multi-product food sources (Table 1). The distribution of sources was organism-dependent; for example, the majority of *Aeromonas* spp. and *Vibrio* spp. isolate sequences were from fish/seafood (Table 1). There were relatively few genomes from ESKAPEE species from food sources. For example, there were only 66 genomes from *Enterobacter* spp. and 23 from *Acinetobacter* spp. (Table 1), whereas there were 9660 and 21,905 genomes from clinical samples, respectively. 

### 3.1. Antimicrobial Resistance by Drug Class

Proportions of predicted resistance by antimicrobial class varied by source depending on both genus (Figure 2) and species (Appendix A). For example, elevated proportions of tetracycline resistance were observed in *Clostridium perfringens* but not *C. botulinum*. A large proportion of clinical *Enterococcus faecium* encoded glycopeptide resistance (86%), compared to only approximately 37% of *Enterococcus faecalis*. Similarly, trimethoprim resistance was predicted for >96% of clinical *Shigella sonnei* compared to <80% in other *Shigella* and *Escherichia* species. Half (approx. 50%) of clinical *Vibrio cholerae* samples encoded aminoglycoside and/or sulphonamide resistance compared to 0.1% of clinical *V. parahaemolyticus* (Appendix A).

Significantly different proportions of predicted resistance between sources were observed for almost all genera, except *Shewanella*, in at least one antimicrobial class (Appendix A). In *Salmonella* and *Campylobacter* species, aminoglycoside resistance was significantly higher in isolates from meat/poultry sources in comparison to clinical and some other food sources (Appendix A). Macrolide resistance in *Bacillus* spp. Was significantly higher in clinical isolates compared to dairy, fruit/vegetable, and multi-product food sources. In *Clostridium* spp., both macrolide and tetracycline resistance were significantly higher in clinical compared to most food sources (Appendix A). *Vibrio* spp. from fish/seafood and multi-product sources had significantly higher proportions of tetracycline resistance compared to clinical, but trimethoprim resistance was significantly lower in fish/seafood compared to most other food sources.

In *Bacillus*, *Escherichia*, and *Klebsiella* species, clinical isolates exhibited a significantly higher prevalence of genes responsible for biocide resistance compared to those found in the majority of food sources. Conversely, predicted biocide resistance in *Listeria* and *Vibrio* was significantly lower in clinical isolates compared to most food sources (Appendix A). Similarly, the prevalence of genes encoding metal resistance in *Listeria* was significantly lower in fruit/vegetable and clinical sources compared to egg, dairy, fish/seafood, meat/poultry, and multi-product sources, and metal resistance in *Vibrio* was significantly lower in clinical isolates compared to fish/seafood and multi-product sources (Appendix A).

For each genus and select species, the association of the resistance type/class with isolation source was investigated using Chi square analyses. Pearson standardized residuals were plotted to measure the strength and direction of the association of a resistance class with a particular source (Appendix A). Notably, correlations between resistance and specific isolation sources were observed for some common foodborne bacterial genera (Table 3, Appendix A).

### 3.2. Antibiotic Resistance

The relative proportion of organisms predicted to be resistant to antibiotic classes varied according to the genera and source of the bacterial isolates (Figure 2). Resistance to antibiotics was frequently significantly higher in human clinical isolates relative to isolates from food sources (Appendix A). For example, sulphonamide resistance was significantly higher in clinical isolates of *Klebsiella* spp. compared to isolates from other sources. Aminoglycoside resistance was significantly associated with clinical *Escherichia* spp. compared to isolates from all other sources. Conversely, resistance to some classes of antimicrobials was significantly higher in meat/poultry isolates (Figure 2, Appendix A). For example, *Salmonella* spp. recovered from meat/poultry were more frequently resistant to aminoglycosides, fosfomycin, sulphonamides, and tetracycline relative to clinical isolates. Note that in some cases, results were biased due to the limited availability of isolates from certain sources. For example, 100% of *Enterobacter* spp. from dairy encoded resistance genes for sulphonamides, yet only two isolate sequences were available in the dataset (Figure 2, Table 1). Additionally, certain species have intrinsic resistance to some antibiotic classes. For example, many of the *Enterobacteriaceae* encode some form of the chromosomal *ampC* gene, resulting in a higher proportion of resistance for the β-lactam class (Figure 2).

The distribution of select antibiotic resistance genes was also investigated (Figure 3, Figure 4 and Figure 5). The analysis of the β-lactam ARG families individually indicated a reduced overall prevalence for this class of antibiotics for many of the gene families (Figure 3). Elevated levels of β-lactam resistance in *Acinetobacter*, *Aeromonas*, *Enterobacter*, *Citrobacter*, *Pseudomonas*, *Staphylococcus*, and *Vibrio* are often due to chromosomally encoded gene families such as *blaACT* and *blaCARB* and not necessarily clinically important gene families (Figure 3). Clinically relevant gene families such as *blaCTX-M*, *blaKPC*, *blaIMP*, and *blaNDM* were observed at elevated proportions in clinical *Citrobacter*, *Enterobacter*, *Klebsiella*, *Pseudomonas*, *Escherichia*, and *Shigella* (Appendix A). Elevated proportions of *blaCTX-M* were observed in *Klebsiella* and *Shigella* from food; however, there were only six *Shigella* isolates included (Appendix A). The carriage of β-lactam gene families also varied by species; for example, approximately 50% of *K. pneumoniae* from clinical, fish/seafood, and fruit/vegetable sources encoded *blaCTX-M* compared to lower levels in other *Klebsiella* spp. from foods; furthermore, approximately 15% of *V. parahaemolyticus* from multi-product food sources encoded *blaCTX-M* and/or *blaCMY* gene families compared to other *Vibrio* species (Appendix A).

Pathogenic species such as *E. coli* and *S. enterica*, targeted by regulatory food-testing programs, were more likely to have ARGs for β-lactams (Figure 3), quinolone (Figure 4), and polymyxin (Figure 5) in comparison to other genera. In *Enterococcus* and *Escherichia* species, significantly higher proportions of quinolone resistance in clinical isolates were due to the carriage of *gyrA*, *parC*, or *parE* mutations (Figure 4, Appendix A).

The *mcr* genes conferring resistance to polymyxins (colistin) were not frequently observed in the genomes investigated (Figure 5). They were most frequently identified in *Aeromonas*, *Enterobacter*, and *Kluyvera* (Figure 5). However, if the *mcr-9* genes were excluded from the analysis, this predicted resistance was much lower for many species.

Vancomycin resistance in *Clostridioides difficile* was due to the carriage of the vanG cluster and was higher in clinical isolates than most food sources (Appendix A). Similarly, in *Enterococcus*, vancomycin resistance was much more prevalent in clinical isolates which encoded either the *vanA* or *vanB* cluster of genes. However, rates of carriage were species specific, with clinical *E. faecium* exhibiting higher rates of carriage than *E. faecalis.* In contrast, vancomycin resistance genes in *Bacillus cereus* were the *vanR*-A/*vanS*-Pt cluster, which were more prevalent in fruit/vegetable, meat/poultry, fish/seafood, and multi-product sources compared to clinical sources (Appendix A).

### 3.3. Biocide Resistance

Similar to antibiotic resistance genes, the presence of biocide resistance genes also varied based on the genus and the source of the isolate. Significantly elevated proportions of isolates carrying the *bcrABC* resistance genes were observed in *Listeria* from food sources compared to clinical sources (Figure 6, Appendix A). Similarly, *qac* resistance genes were more prevalent in *Vibrio* spp. isolated from food sources compared to clinical (Figure 6). In contrast, biocide resistance was significantly higher in clinical *Escherichia* compared to all food sources except egg (Figure 6, Appendix A). As with antibiotic resistance, certain bacteria encode biocide resistance determinants chromosomally. For example, most *Staphylococcus* [species] encode *lmrS*, and a chromosomal *emrE* is found in most *Klebsiella*.

We investigated the potential co-carriage of biocide and antimicrobial resistance; however, the limited availability of isolates from some sources hindered the determination of the significance of the associations (Appendix A). Of note, similar proportions of antibiotic resistance and AMR + Biocide resistance were observed for the following: sulphonamide and trimethoprim in *Klebsiella* from meat/poultry and multi-product sources; beta-lactam, quinolone, sulphonamide, tetracycline, and trimethoprim in *Escherichia* from egg and fish/seafood sources. These results suggest potential co-carriage in these genera from these sources (Appendix A).

### 3.4. Metal Resistance

The presence of metal resistance genes varied among genera and the source of isolation (Figure 7). *Listeria* spp. generally carried few genes predicted to confer resistance to metals, with *cad* and *ars*, encoding resistance to cadmium and arsenite, being the most common (Figure 7). Almost all *Salmonella* sequences encoded the *gol* gold resistance gene, and *Salmonella* spp. also had higher proportions of arsenite resistance determinants in food isolates compared to clinical isolates. Additionally, silver resistance was higher in *Salmonella* isolates from meat/poultry and multi-product food sources compared to clinical sources. Approximately 34–50% of *Campylobacter* isolates from clinical, meat/poultry, and multi-product sources encoded arsenite *ars* and/or *acr* resistance genes, with a higher proportion of meat/poultry isolates encoding arsenite resistance compared to clinical isolates (Figure 7). In *Klebsiella*, high proportions of resistance to metals were due to the carriage of *ars* (arsenite), *pco* (copper), *mer* (mercury), *sil* (silver), and *ter* (tellurium) (Figure 7). Significantly higher proportions of metal resistance in *Escherichia* from meat/poultry and multi-product foods were due to the carriage of *pco* (copper), *sil* (silver), and *ter* (tellurium) resistance determinants in these sources (Figure 7, Appendix A). Cadmium (*cad*) resistance in *Staphylococcus* spp. was slightly higher in multi-product isolates than clinical; however, *cad* was detected in 56%, 59%, and 64% of meat/poultry, clinical, and multi-product *Staphyloccocus* sequences, respectively.

## 4. Discussion

This study leveraged published bacterial genomes to explore the link between antibiotic resistance genes (ARGs) and bacteria from food and human clinical sources, employing the NPDD as a key tool for this analysis. While this is a valuable resource, the results are subject to certain limitations. Notably, bacterial isolates from food sources are significantly outnumbered by those from human-clinical samples and available data may be biased due to the non-systematic nature of food sampling, including the presence of genomes from clonal isolates. Moreover, the detection of ARGs is contingent on the quality of the genome assemblies, with closed genomes typically enabling more reliable detection of ARGs compared to lower-quality draft genomes [60,61]. As such, caution should be taken for any statistical inferences being deduced from our results, in particular where a low number (<10) of isolates were investigated [62]. Despite these constraints, this study offers an overview of the relationship between ARGs and different food sources and highlights current gaps in the surveillance of agri-food products to monitor the emergence of AMR.

### 4.1. The Importance of Metadata

As genomic sequencing technologies advance and the volume of sequence data increases, adopting standardized methods for metadata collection and reporting is crucial to maximize the impact of large publicly available repositories [63]. Acknowledging this critical need, the ISO 23418:2022 standard for the whole-genome sequencing of bacteria provides extensive guidelines for metadata collection [64]. The NPDD has begun incorporating IFSAC categories [53,54] into its metadata, but updates and manual curation are still needed. As found in other studies, manual curation was needed to resolve issues wherein at least one component of the metadata, such as host or isolation source, was either missing, inconsistent, or misspelled [65,66,67,68]. The curation of metadata after the fact is a daunting task and subject to error, especially in the case of older entries where information may no longer be easily retrievable. Collaborative efforts are ongoing to standardize the collection of metadata [69] and include standardized structured vocabulary derived from specific ontologies including Environmental Ontology (ENVO) and Foundations of Medical Anatomy (FMA) [63,70,71]. More recently, a harmonized food ontology (FoodOn) was developed to address food product terminology gaps [72]. Tools such as METAGENOTE have been developed that facilitate the annotation of sample data prior to uploading sequence files to the SRA [63]. As NCBI Pathogen Detection continues to improve data collection methods and update its current repository with standardized defined ontology, this resource will become even more valuable for conducting large meta-analyses.

### 4.2. Filling the Gaps in Agri-Food Testing and Resistance Surveillance

This study analysed NCBI pathogen data for 639,087 bacterial genomes isolated from clinical (76.88%) and food (23.12%) sources to assess the connection between predicted AMR and food sources (Table 1). Despite inherent data limitations, we observed several associations between ARGs of concern and isolation sources. In general, ARGs were more prevalent in clinical isolates, with a few exceptions. In particular, *Salmonella* isolates from meat/poultry were more likely to harbour ARGs associated with multiple resistance classes including β-lactams, quinolones, sulphonamides, and tetracycline. We found that *B. cereus* from meat/poultry and fruit/vegetable sources were more likely to encode tetracycline resistance and/or the *vanR-A/vanS-Pt* glycopeptide resistance cassette(s) (Appendix A). However, previous studies found although *vanR*-alleles were detected in 100% of *B. cereus* isolates studied, all were susceptible to vancomycin [73]. *Clostridium botulinum* from multi-product and fruit/vegetable sources also had higher rates of carriage for fosfomycin, metal, and phenicol resistance.

Agri-food production practices can impact selection for AMR organisms. For example, recent studies have implicated the use of ceftiofur in poultry production with an increase in third generation cepholosporin (3GC)-resistant *Salmonella* Heidelberg in both poultry and associated with human illness [74,75,76]. The use of antimicrobials in agri-food production has been shown to lead to the co-selection of critically important AMR [77], and subinhibitory concentrations of antimicrobials can increase the dissemination of MGEs harbouring ARGs [78,79]. In food crops, the use of fertilizers from animal or human sources has been associated with an increase in AMR organisms [80]. A recent meta-analysis indicated that between 3.75 and 4.63% of food crops harboured *Enterobacteriaceae* resistant to tetracycline or 3GCs, with prevalences varying by country [80]. The correlation between the antimicrobial resistance of specific classes and certain isolation sources is corroborated by other studies for some genera. For example, Zaheer et al. also reported high levels of tetracycline and macrolide resistance in *Enterococcus* from human clinical and cattle sources and trimethoprim resistance in up to 83% of clinical *Enterococcus faecium* isolates [81].

Another notable finding was the high rates of biocide resistance in *L. monocytogenes* isolates from egg, multi-product, dairy, and meat/poultry sources contrasting with low levels of this resistance in clinical isolates (Figure 2 and Figure 3, Appendix A). A previous analysis of 1279 *L. monocytogenes* strains from food products found that five of the most frequently isolated clonal complexes (CCs) of *L. monocytogenes* were significantly more likely to encode gene(s) conferring biocide resistance [82]. In almost all resistant isolates, QAC resistance was plasmid-borne, suggesting that the transfer of plasmid-borne sanitizer resistance may be associated with pathogen persistence in food production.

*Listeria* spp. from food sources also encoded arsenite and cadmium resistance determinants at higher rates than clinical isolates (Figure 7). Resistance to cadmium and arsenic is one of the earliest documented metal resistance phenotypes of *L. monocytogenes*. Arsenic has been primarily associated with serotype 4b (over-represented clinical type), and arsenic resistance is most frequently encountered among clones associated with outbreaks [83].

Not all AMR organisms or ARGs are of equal importance to the current AMR crisis [84,85]. The WHO and CDC list carbapenem- and ESBL-producing *Enterobacteriaceae* as a critical priority and drug-resistant *Salmonella*, *Campylobacter*, *E. faecium*, *S. aureus*, and *Helicobacter pylori* as high-priority pathogens [86,87]. While foods are currently monitored for the presence of pathogens such as Shiga toxin-producing *E. coli* (STEC), *Vibrio*., *Salmonella*, and *Campylobacter* species, there is limited surveillance on the critical- and high-priority ESKAPEE pathogens in foods, despite evidence indicating that these species are commonly found in foods [88,89]. Of these species, *S. aureus* and *E. coli* had the highest representation in the NPDD (Table 1, Appendix A).

The prevalence of *mec* encoded β-lactam resistance in *S. aureus* did not significantly differ between both meat/poultry and multi-product sources relative to clinical sources, whereas this resistance was rare in isolates from dairy sources *(*Figure 2 and Figure 3, Appendix A). The *mec* genes are found in methicillin-resistant *S. aureus* (MRSA) strains, which are a global concern and were estimated to be responsible for 100,000 deaths in 2019 [20]. A study by Bouchami et al. [90] on the pork production chain found MRSA to be present in live pigs, meat, the slaughterhouse environment, and workers, with 55% encoding the *mec* cassette and 61% encoding the biocide resistance gene *lmrS*. Interestingly, our study carriage of *mec* was significantly lower in dairy compared to all other foods and clinical sources (Figure 3, Appendix A). This finding is similar to a meta-analysis conducted by Khanal et al. [91], who reported the prevalence of MRSA to be 3.81% overall and 3.91% in dairy cattle farms and cattle milk specifically. While MRSA isolates are commonly recovered from foods, the role of food in their transmission remains unclear [92].

Despite the under-representation of certain species in foods (e.g., only 23 *Acinetobacter* from food and over 20,000 from clinical samples, Table 1), we found carbapenem and ESBL resistance gene family ESKAPEE pathogens in foods including dairy, egg, fish/seafood, fruit/vegetable, meat/poultry, multi-product, spice/herbs, and nuts/seeds sources (Appendix A). The CTX-M family is the most prevalent type of ESBL observed in ESBL-producing *Enterobacteriaceae* found in vegetables [93]. We found a similar prevalence of *bla*_CTX-M_-encoding *K*. *pneumoniae* from both clinical and fruit/vegetable sources (Appendix A). Other studies also reported high levels of *bla*_CTX-M_-encoding *K. pneumoniae* isolated from fruit/vegetable sources [93]. *P. aeruginosa* from fruit/vegetable sources encoded *bla*_VIM_ at a higher rate than clinical isolates. To elucidate the significance of the role of food in the transmission of ESKAPEE pathogens, larger scale, targeted studies are needed to address current data gaps.

Note that the presence of β-lactam genes does not necessarily correlate with the production of ESBLs. For example, resistance to penicillin and 1st- and 2nd-generation cephalosporins is often mediated by chromosomal β-lactamase genes, such as *ampC* and *ampC*-type genes. These genes are often species specific (*Acinetobacter*, *bla*_OXA_; Citrobacter, *bla*_CMY_; *Enterobacter*, *bla*_ACT_, *bla*_ADC_) and alleles often do not, but in some situations may, confer resistance to 3rd- and 4th-generation cephalosporins or carbapenems (Figure 2 and Figure 3). Additionally, resistance to some antimicrobials may be conferred by single nucleotide variants (SNVs) in the bacterial genome, of which only a few are well characterized. Given their location on the chromosome, both of these gene types present a lower risk of transmission [20,92,94].

While the NPDD currently offers limited data on ARG-encoding foodborne bacteria, this absence does not necessarily imply that these organisms are absent in foods. The availability of data regarding bacterial abundance in food sources is often constrained to focused studies targeting specific commodities, genera, or species and is heavily influenced by factors such as the targeted bacteria, location, and seasonal variations. For instance, a 2013 survey conducted in Canada on fresh fruits and vegetables reported generally a very low prevalence of bacterial pathogens such as *Salmonella*, *E. coli* O157, *Shigella*, *Campylobacter*, and *L. monocytogenes* [95]. In contrast, a recent study exploring ready-to-eat foods, including meat products as well as fruit, in developing countries reported a prevalence range from 6.1–34.4% for many ESKAPEE pathogens, *Salmonella*, *Bacillus*, and *Shigella*, depending on the organism [96]. 

However, there remains relatively few studies investigating or reporting the prevalence of typically non-targeted foodborne genera and species such as *Citrobacter* and non-*pneumoniae Klebsiella* species This study highlights that AMR of concern is present in ESKAPEE pathogens isolated from food sources and that we often observe clinical priority ARGs in these species. Nonetheless, the available data are highly biased towards clinical sources. Although some studies have reported the presence of multidrug-resistant pathogens from foods such as fruit and vegetables, the body of research in this area is still relatively small. Few studies quantify the risk associated with consumption, and many focus exclusively on specific bacterial pathogens [97]. Additionally, certain emerging high-priority resistance genes are rarely found, even in clinical isolates (Appendix A). Given the existing gaps in data from food sources, it is difficult to measure transmission from these to clinical settings. More targeted surveillance is needed to ascertain whether foods are a risk source and potential transmission route for AMR [98].

Understanding the interplay between resistance and MGEs is critical for understanding the spread and dissemination of ARGs across bacterial populations and environments. While linking ARGs with MGEs is crucial for assessing the transmissibility of AMR, the utilization of metadata in our study precluded the precise association of resistance genes with specific MGEs. Bioinformatic tools, such as mob-suite, permit the reconstruction of plasmids using isolate genome sequence data [99]. Furthermore, an important AMR resource, the Comprehensive Antibiotic Resistance Database (CARD), now integrates information on the presence of ARGs and their corresponding plasmid location(s) derived from the analysis of NCBI whole-genome sequence data [100,101]. However, unlike our study the data in CARD are not categorized by isolation source. The surveillance of AMR in agri-food samples may benefit by shifting focus from the isolation/testing of specific organisms to investigating MGEs being transferred throughout food production (e.g., plasmidome sequencing) [102]. These elements provide a mechanism to distribute genes that are beneficial for survival and often carry genes encoding virulence factors; antibiotic-, biocide-, and metal-resistance; and functions involved in host–bacterial interactions [103]. Therefore, MGEs may contain resistance genes of the highest risk and clinical relevance in agri-food production samples.

## 5. Conclusions

Food products, facilities, and food-producing animals contain a variety of bacteria, and antimicrobial use in agriculture is an alleged driver for increasing AMR [98]. Current monitoring programs target select bacterial pathogens within products (e.g., *Salmonella* spp., STEC, *Vibrio* spp., and *L. monocytogenes*, among others). As species that are of concern for AMR, such as the ESKAPEE pathogens, are not routinely investigated, the AMR burden of foods remains unclear. This study illustrates how high-quality, publicly available bacterial genome sequences can provide insights on the distribution of ARGs in agri-food production. In comparison to foodborne pathogenic species, there was relatively limited coverage of ESKAPEE species recovered from food sources in the NPDD, despite their importance in human infection. However, these data still provide an overview of the types of ARGs in bacteria isolated from food and clinical sources.

As samples found throughout the food production continuum are often compositionally complex, methods that will enable the evaluation of the resistance burden in the food chain are required. These methods should target bacteria that may serve as reservoirs for ARGs in food production. Additional sequence data generation for AMR in ESKAPEE pathogens such as *Enterobacter* and *Klebsiella*, including bacteria from lesser studied food sources, is essential for evaluating the resistance burden in food production. 

## Figures and Tables

**Figure 1 microorganisms-12-00709-f001:**
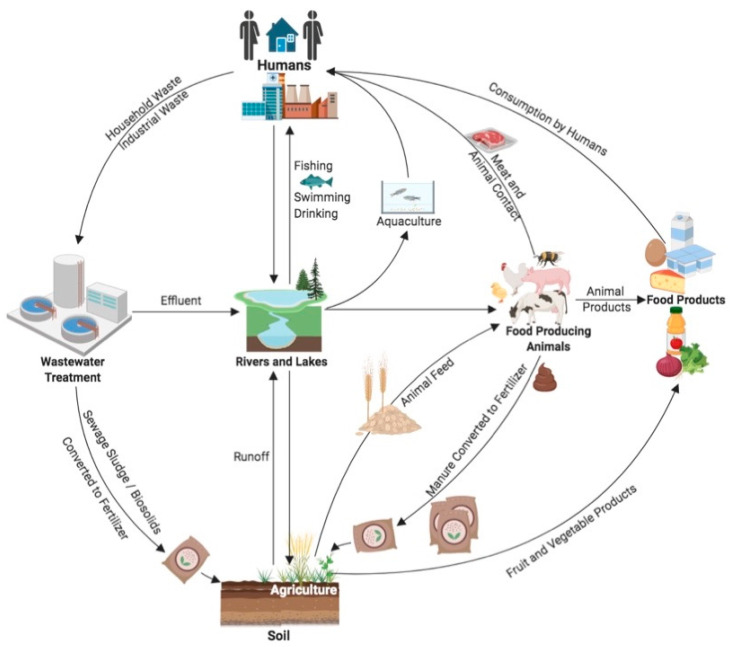
Potential routes of transmission of bacteria and ARGs through the environment and food production systems. Arrows indicate routes of dissemination among different environments. Humans represents all human-related activities including clinical, industrial, and household. Intricacies of food production processes including processing, pasteurization, slaughter, sanitization, packaging, preparation, etc., are not displayed but are inferred by arrows from agriculture to food products, from aquaculture to humans, and from animals to both food products and humans. (Figure created using BioRender.com accessed on 8 December 2023).

**Figure 2 microorganisms-12-00709-f002:**
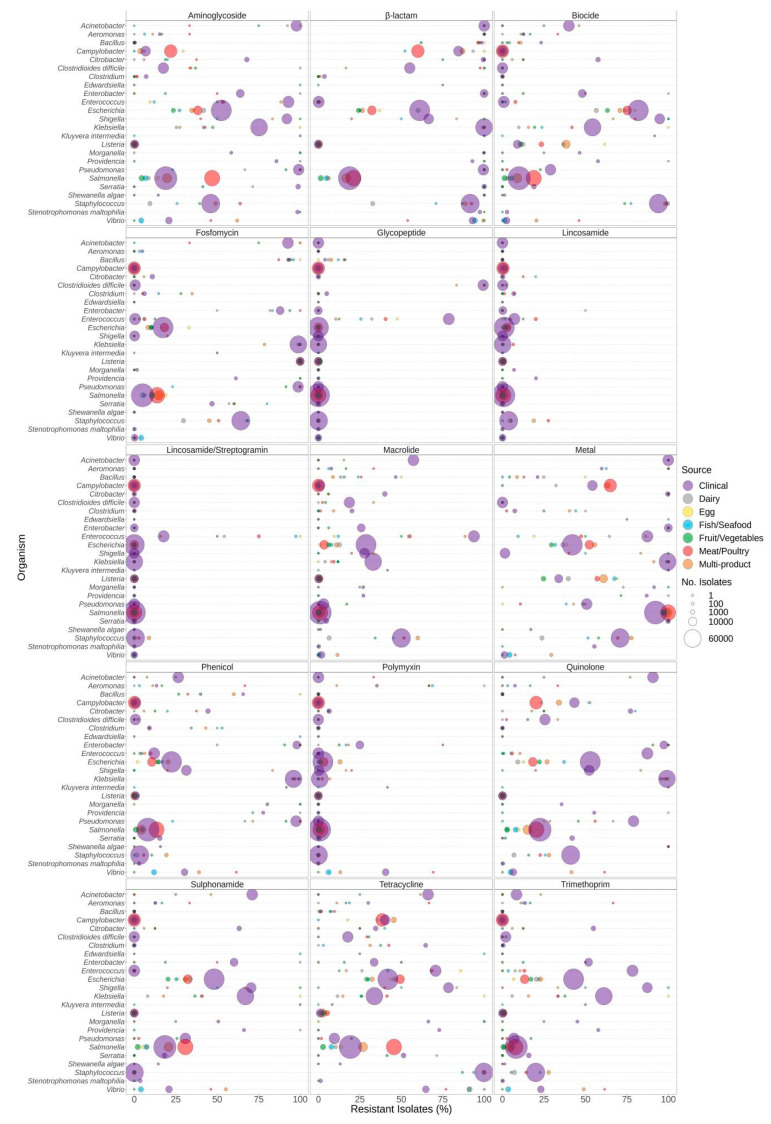
Predicted resistance to antimicrobial classes in 639,087 food and human clinical bacterial genomes published in the NCBI Pathogen Detection database. Presence of ARGs and source of bacterial isolates was determined based on metadata files associated with the whole-genome sequences published in the NCBI Pathogen Detection database. For each organism listed (*y*-axis), the percentage (*x*-axis) of isolates from each source (see colour legend) predicted to be resistant to classes of antimicrobials (panel headings) is displayed. Bubble diameters correspond to the total number of isolates with predicted resistance from each source (no. of isolates). Note that the quinolone class includes both acquired AMR genes (e.g., *qnrS*) and chromosomal point mutations (*gyrA*, *parE*, *parC*) reported for only some of the genera.

**Figure 3 microorganisms-12-00709-f003:**
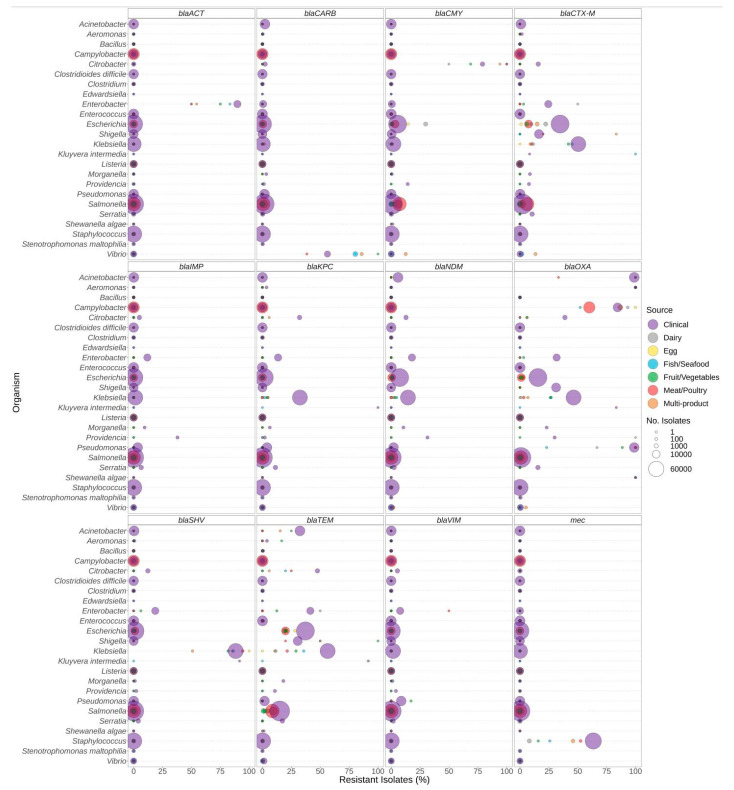
ß-lactam resistance genes observed in bacteria commonly found in food products as a function of isolation source. Presence of ARGs and source of bacterial isolates was determined based on metadata files associated with the whole-genome sequences published in the NCBI Pathogen Detection database (*n* = 639,087). For each genus or species listed (*y*-axis), the percentage (*x*-axis) of isolates from each source (see colour legend) carrying a ß-lactam resistance gene (panel headings) is displayed. Bubble diameters correspond to the total number of isolates with the resistance gene from each source (no. of isolates). For each gene, all alleles in the AMRFinderPlus database are included. Most gene families displayed include alleles conferring priority (or critical) resistance, except for *bla*_ACT_ and *bla_CARB_*, which are often chromosomally encoded by *Enterobacter* and *Vibrio* species.

**Figure 4 microorganisms-12-00709-f004:**
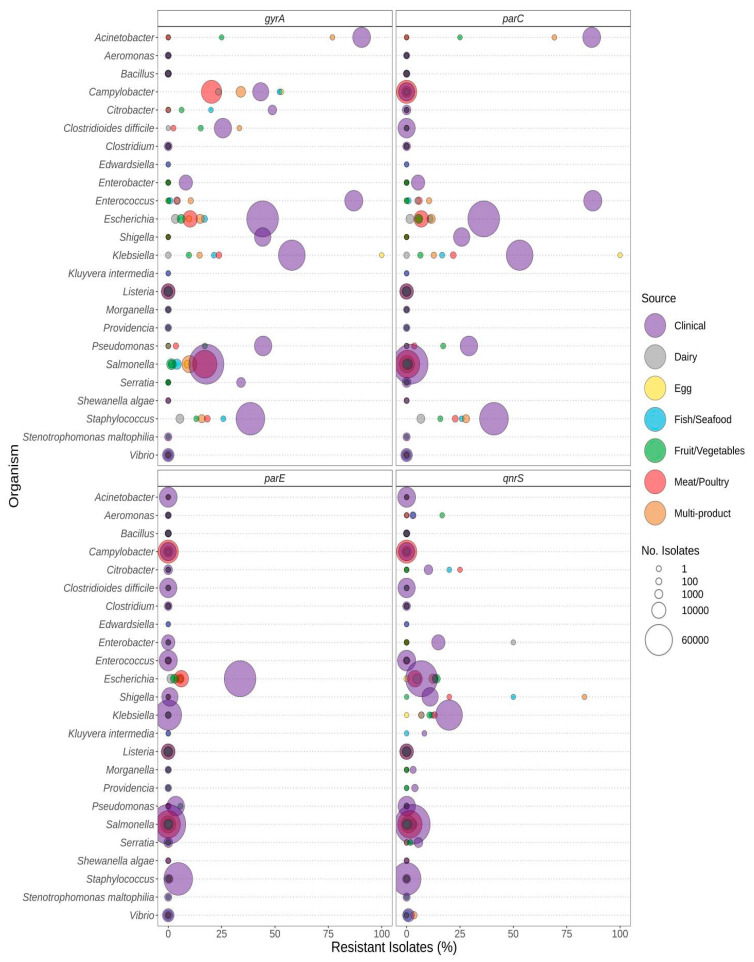
Quinolone resistance observed in bacteria commonly found in food products as a function of isolation source. Presence of ARGs and source of bacterial isolates was determined based on metadata files associated with the whole-genome sequences published in the NCBI Pathogen Detection database (*n* = 639,087). For each genus or species listed (*y*-axis), the percentage (*x*-axis) of isolates from each source (see colour legend) with a quinolone-resistance gene (panel headings) is displayed. Bubble diameters correspond to the total number of isolates with the resistance gene from each source (no. of isolates). Note that the analyses of point mutations in *gyrA*, *parC*, and *parE* conferring quinolone resistance are not available for all species (i.e., mutations may be present in some genera but not reported in this study).

**Figure 5 microorganisms-12-00709-f005:**
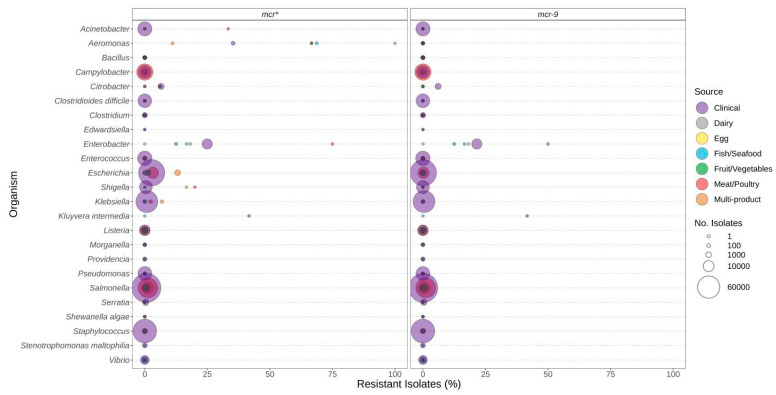
Polymyxin (e.g., colistin) resistance genes observed in bacteria commonly found in food products as a function of isolation source. Presence of ARGs and source of bacterial isolates was determined based on metadata files associated with the whole-genome sequences published in the NCBI Pathogen Detection database (*n* = 639,087). For each genus or species listed (*y*-axis), the percentage (*x*-axis) of isolates from each source (see colour legend) with each antibiotic resistance gene (panel headings) is displayed. Bubble diameters correspond to the total number of isolates with the resistance gene from each source (no. of isolates). The *mcr** panel includes all *mcr*-alleles (1 through 10), including those in the *mcr-9* panel.

**Figure 6 microorganisms-12-00709-f006:**
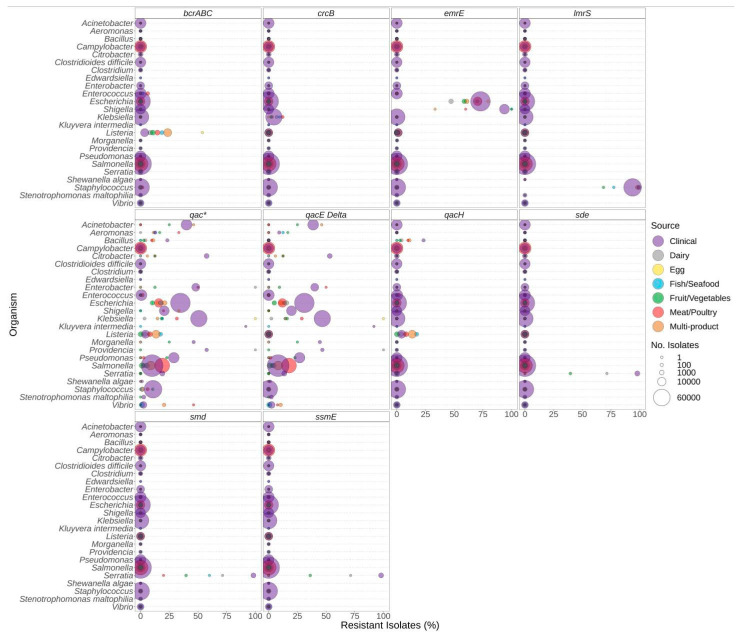
Biocide resistance genes observed in bacteria commonly found in food products as a function of isolation source. Presence of biocide resistance genes and source of bacterial isolates was determined based on metadata files associated with the whole-genome sequences published in the NCBI Pathogen Detection database (*n* = 639,087). For each genus or species listed (*y*-axis), the percentage (*x*-axis) of isolates from each source (see colour legend) with each biocide resistance gene (panel headings) is displayed. Bubble diameters correspond to the total number of isolates with the resistance gene from each source (no. of isolates). The *qac** panel includes data for all *qac*-alleles, including those in other panels.

**Figure 7 microorganisms-12-00709-f007:**
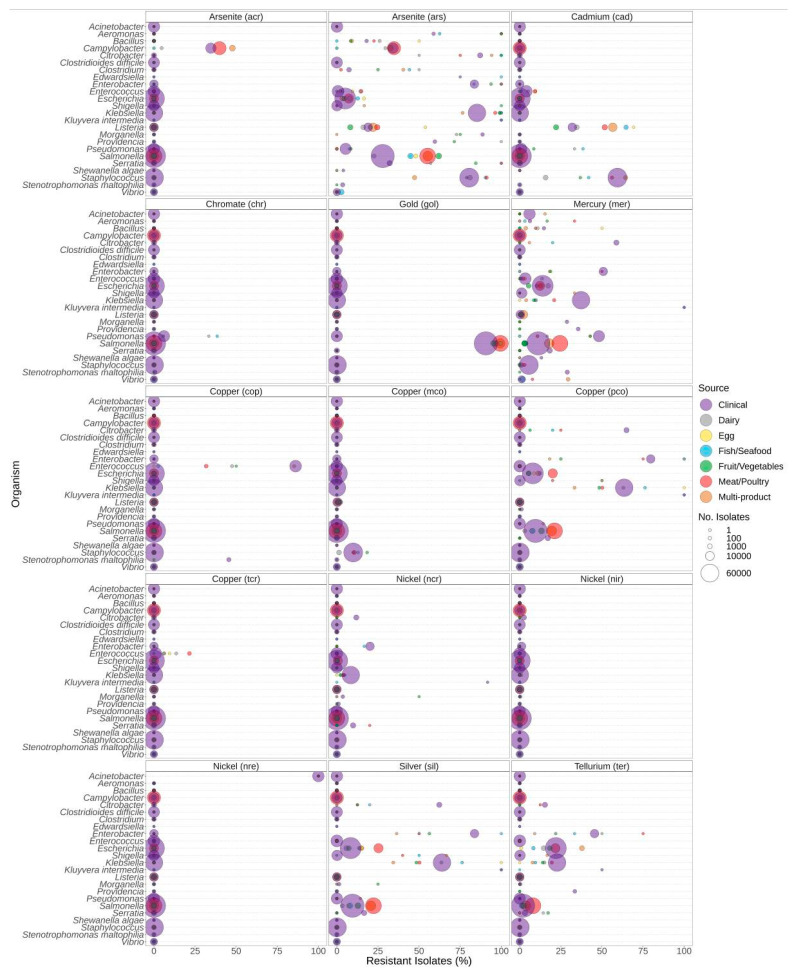
Metal resistance genes observed in bacteria commonly found in food products as a function of isolation source. For each genus or species listed on the *y*-axis, the percentage of isolates from each source (see colour legend) with each respective metal resistance gene (panel headings) is displayed (*x*-axis). Panel headings indicate the predicted metal that indicated gene (in parentheses) confers resistance to. Sizing of points corresponds to the total number of isolates of that genus for isolation source (no. of isolates).

**Table 1 microorganisms-12-00709-t001:** Total number of genome assemblies from food and human clinical sources analysed for each organism.

Organism ^a^	Number of Sequences from Food Source (%) ^b^											Total from Foods (%) ^d^	Human Clinical (%) ^e^
	Egg	Fish/ Seafood	Multi-Product ^c^	Meat/Poultry	Cider	Dairy	Flour	Fruit/ Vegetable	Spice/ Herbs	Nuts/Seeds	Tea		
*Acinetobacter*	2 (8.70%)	1 (4.35%)	13 (56.52%)	3 (13.04%)	-	-	-	4 (17.39%)	-	-	-	23 (0.02%)	21,905 (4.46%)
*Aeromonas*	-	32 (59.26%)	9 (16.67%)	6 (11.11%)	-	1 (1.85%)	-	6 (11.11%)	-	-	-	54 (0.04%)	306 (0.06%)
*Bacillus*	2 (0.28%)	25 (3.44%)	410 (56.40%)	31 (4.26%)	-	165 (22.70%)	-	71 (9.77%)	17 (2.34%)	6 (0.83%)	-	727 (0.49%)	326 (0.07%)
*Campylobacter*	17 (0.05%)	23 (0.07%)	3045 (8.95%)	30,807 (90.50%)	-	149 (0.44%)	-	-	-	-	-	34,041 (23.03%)	16,577 (3.37%)
*Citrobacter*	-	5 (10.20%)	17 (34.69%)	8 (16.33%)	-	2 (4.08%)	-	16 (32.65%)	1 (2.04%)	-	-	49 (0.03%)	2013 (0.41%)
*Clostridioides difficile*	-	-	6 (6.38%)	41 (43.62%)	-	1 (1.06%)	-	46 (48.94%)	-	-	-	94 (0.06%)	20,105 (4.09%)
*Clostridium*	-	34 (12.98%)	109 (41.60%)	80 (30.53%)	-	2 (0.76%)	-	32 (12.21%)	4 (1.53%)	1 (0.38%)	-	262 (0.18%)	1268 (0.26%)
*Edwardsiella*	-	3 (100.00%)	-	-	-	-	-	-	-	-	-	3 (<0.00%)	4 (<0.00%)
*Enterobacter*	-	6 (9.09%)	11 (16.67%)	4 (6.06%)	-	2 (3.03%)	-	32 (48.48%)	4 (6.06%)	7 (10.6%)	-	66 (0.04%)	9660 (1.97%)
*Enterococcus*	21 (2.34%)	117 (13.01%)	95 (10.57%)	538 (59.84%)	-	118 (13.13%)	-	8 (0.89%)	1 (0.11%)	1 (0.11%)	-	899 (0.61%)	23,002 (4.68%)
*Escherichia*	103 (0.59%)	337 (1.93%)	1541 (8.84%)	12,399 (71.15%)	9 (0.05%)	1408 (8.08%)	93 (0.53%)	1373 (7.88%)	147 (0.84%)	16 (0.09%)	-	17,426 (11.79%)	90,808 (18.48%)
*Klebsiella*	1 (0.16%)	42 (6.69%)	102 (16.24%)	215 (34.24%)	-	173 (27.55%)	-	93 (14.81%)	2 (0.32%)	-	-	628 (0.42%)	60,726 (12.36%)
*Kluyvera intermedia*	-	1 (100.00%)	-	-	-	-	-	-	-	-	-	1 (<0.00%)	12 (<0.00%)
*Listeria*	13 (0.08%)	933 (5.59%)	9783 (58.58%)	2303 (13.79%)	-	1488 (8.91%)	-	2090 (12.52%)	53 (0.32%)	36 (0.22%)	-	16,699 (11.30%)	10,218 (2.08%)
*Morganella*	-	-	2 (11.76%)	-	-	11 (64.71%)	-	4 (23.53%)	-	-	-	17 (0.01%)	359 (0.07%)
*Providencia*	-	-	-	-	-	1 (12.50%)	-	7 (87.50%)	-	-	-	8 (0.01%)	490 (0.10%)
*Pseudomonas*	4 (2.25%)	13 (7.30%)	7 (3.93%)	111 (62.36%)	-	6 (3.37%)	-	35 (19.66%)	-	2 (1.12%)	-	178 (0.12%)	20,547 (4.18%)
*Salmonella*	622 (0.88%)	2247 (3.18%)	13,375 (18.90%)	48,436 (68.45%)	-	622 (0.88%)	4 (0.01%)	2933 (4.14%)	831 (1.17%)	1688 (2.39%)	2 (<0.0%)	70,760 (47.88%)	114,170 (23.24%)
*Serratia*	-	5 (6.58%)	-	5 (6.58%)	-	7 (9.21%)	-	58 (76.32%)	-	1 (1.32%)	-	76 (0.05%)	2167 (0.44%)
*Shewanella algae*	-	4 (80.00%)	1 (20.00%)	-	-	-	-	-	-	-	-	5 (<0.00%)	76 (0.02%)
*Shigella*	-	2 (14.29%)	6 (42.86%)	5 (35.71%)	-	-	-	1 (7.14%)	-	-	-	14 (0.01%)	17,281 (3.52%)
*Staphylococcus*	-	31 (1.18%)	891 (33.89%)	246 (9.36%)	-	1408 (53.56%)	15 (0.57%)	38 (1.45%)	-	-	-	2629 (1.78%)	72,276 (14.71%)
*Stenotrophomonas maltophilia*	-	-	-	-	-	-	-	1 (100.00%)	-	-	-	1 (<0.00%)	872 (0.18%)
*Vibrio*	-	2642 (84.46%)	469 (14.99%)	13 (0.42%)	-	-	-	2 (0.06%)	2 (0.06%)	-	-	3128 (2.12%)	6131 (1.25%)
Combined total from source ^f^	785 (0.53%)	6503 (4.40%)	29,892 (20.23%)	95,251 (64.45%)	9 (0.01%)	5564 (3.76%)	112 (0.08%)	6850 (4.64%)	1062 (0.72%)	1758 (1.19%)	2 (<0.0%)	147,788	491,299

^a^ Where only genus is listed, multiple species for corresponding genus were included in analysis. ^b^ Number of sequences investigated from corresponding food source (column headers) for organism listed (row name). The percentage of sequences from corresponding food source, out of the “total from foods” for that organism, is in parentheses. ^c^ Refers to mixed food products that could not be easily placed in a single category (e.g., meat and cheese sandwich, macaroni salad, brownie, etc.). ^d^ The total number of sequences from food sources for corresponding organism. The proportion out of the total number of food isolates (147,788) that an organism constitutes is in parentheses (%). ^e^ Clinical sources only included isolate sequence submissions with the epi_type listed as “clinical” and host as *Homo sapiens*. The proportion, out of the total number of clinical isolates (491,299), that an organism constitutes is in parentheses (%). ^f^ The combined total number of all organisms investigated from that particular food source. The percentage of sequences from that food source, out of all food sources, is in parentheses. For the total from food and human clinical sources (last two columns), only the total number of sequences investigated from each source is listed.

**Table 2 microorganisms-12-00709-t002:** Food isolation source definitions.

Isolation Source Assignment *	Definition	Examples
Dairy	Dairy products including milk, ice cream, and cheeses. Milk from bovine with mastitis was excluded.	Milk from healthy cattle, raw milk, Roquefort papillon cheese, etc.
Egg	Egg products such as chicken eggs and chicken egg shells but not including reptile or fish eggs	Chicken egg outside shell, frozen liquid egg, egg white, yolks, etc.
Fish/Seafood	Fish and seafood products, excluding mixed salads and mixed products, which were categorized as multi-product.	Brown mussels, imported shrimp, salmon, crab, etc.
Fruit/Vegetables	Any fruit or vegetables, including frozen and ready to eat, and mixed fruit sources. French fries listed as multi-product.	Tomato, red leaf lettuce, carrot, mango.
Multi-product	Mixed food products or products that cannot be easily categorized. Chili, if type was not specified, as it could refer to prepared chili or the pepper; spreads and cream cheese mixtures; all salads (including tuna, egg, potato, and coleslaw) that may contain mixed ingredients; hummus; guacamole; salsa; ready-to-eat mixed products; sandwiches; fruitcake; sushi; pasta; sauces; etc.	Tuna salad, meatball sub, brownie, coleslaw, pie crust, smoothie blend, etc.
Meat/Poultry	Meat and poultry products including raw and ready to eat products, sausages, hot dogs, snails, etc. but excluding reptile meats and mixed products (like meat sauce, pates, and spreads)	Packaged whole turkey, thin sliced chicken breast, venison, raw beef, beef trim, etc.

* Only food isolation sources investigated and discussed in this publication are described. Definitions for additional sources, including the dictionary file containing all sources, are available at https://github.com/OLC-Bioinformatics/source_and_resistance_categorizer.git, last updated November 2023 under “Source Definitions”. Additional sources may require further curation.

**Table 3 microorganisms-12-00709-t003:** Associations of antimicrobial resistance with specific food isolation sources.

Genus	Resistance Class(es) with Positive Association to Source	Source(s)
*Bacillus*	Glycopeptide	Fruit/Vegetables
*Campylobacter*	Aminoglycoside	Meat/poultry, Egg
	Metal, Tetracycline	Clinical, Dairy, Meat/Poultry, Multi-product
*Citrobacter*	Biocide, Sulphonamide, Trimethoprim	Clinical
*Clostridium*	Macrolide	Clinical
	Metal, Phenicol	Multi-product, Fruit/Vegetables, Fish/Seafood, Dairy
	Tetracycline	Clinical, Meat/Poultry, Multi-product, Fruit/Vegetables
*C. difficile*	Glycopeptide	Clinical, Meat/Poultry, Multi-product
*Enterococcus*	Glycopeptide, Quinolone, Trimethoprim	Clinical
*Escherichia*	Trimethoprim	Clinical (weak association)
*Shigella*	Trimethoprim	Clinical (very strong association)
*Klebsiella*	Beta-lactam, Metal, Phenicol, Quinolone	Clinical, Meat/Poultry, Multi-product, Fruit/Vegetables, Fish/Seafood, Dairy, Egg
	Sulphonamide, Trimethoprim	Clinical, Egg
*Listeria*	Biocide	Multi-product, Egg
*Salmonella*	Aminoglycoside, Tetracycline	Meat/Poultry
*Vibrio*	Tetracycline	All sources, but especially strong with Fish/Seafood
	Aminoglycoside, Sulphonamide, Trimethoprim	Multi-product

Abbreviations: *C. difficile*: *Clostridioides difficile*.

## Data Availability

The data presented in this study are available in the NCBI Pathogen Detection database (NPDD) at https://www.ncbi.nlm.nih.gov/pathogens/ last accessed 17 November 2023. Detailed descriptions of the metadata files used from the NPDD are available in Appendix A. The code and dictionary files used to annotate and analyse NPDD data tables are openly available on GitHub at https://github.com/OLC-Bioinformatics/source_and_resistance_categorizer last updated November 2023.

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
