# Peer review of "Analysis of Antimicrobial Resistance in Bacterial Pathogens Recovered from Food and Human Sources: Insights from 639,087 Bacterial Whole-Genome Sequences in the NCBI Pathogen Detection Database"

_microorganisms, 2024, doi:10.3390/microorganisms12040709_

Round 1
Reviewer 1 Report
Comments and Suggestions for Authors
This study utilizes the NCBI Pathogen Detection database to conduct a comprehensive analysis of antibiotic resistance genes in bacterial species isolated from foodborne and human clinical sources. It highlights the current gaps in the surveillance of agri-food products that make it difficult to monitor the emergence of antimicrobial resistance. The authors also note that this study may not fully capture the diversity and prevalence of resistance genes among commensal foodborne bacteria due to potential biases and limitations in the database, which may affect the analysis and interpretation of the findings. Overall, the manuscript is well-written with sound knowledge and solid data, and may be suitable for publication in its current form.
Reviewer 2 Report
Comments and Suggestions for Authors
Documenting the presence of antibiotic resistance genes in bacterial species is a required first step in understanding the emergence and spread of antimicrobial resistance. Food products, facilities, and food-producing animals contain a variety of bacteria, and antimicrobial use in agriculture is an alleged driver for increasing AMR. This study illustrates how high-quality, publicly-available bacterial genome sequences can provide insights on the distribution of ARGs in agri-food production. In comparison to foodborne pathogenic species, there was relatively limited coverage of ESKAPEE species recovered from food sources in the NPDD, despite their importance in human infection. The obtained by authors data still provides an overview of the types of ARGs in bacteria isolated from food and clinical sources. However, the issue of ARG of pathogenic and opportunistic bacteria is not new and it has been widely studied, but it is not mentioned by the authors themselves.
Despite this, the results are quite interesting but still the flaws are needed to be addressed. Therefore, the paper needs major revision - specific comments:
1. The importance of studying ARGs in any kind of biological systems, is always well-described. However, there are not enough evidence or justify, why the authors needed performed this study? I think in the introduction part; the actuality of this investigation should be stronger justified.
2. As authors said themselves, The NCBI Pathogen Detection database (NPDD) resource “integrates bacterial and fungal pathogen genomic sequences from numerous ongoing surveillance and research efforts”, and includes data from food, environmental, and clinical sources of both human and animal origin. The reason why authors chose to study only bacterial genomes? What about fungal pathogens and are they a small part of NPDD? It should be mentioned.
3. How were genomes downloaded? Using the NCBI Datasets command-line tools (CLI) v13.x? If so, it should be mentioned.
4. Table 1. Total number of sequences of each organism analysed from food and human clinical sources. Number of sequences of what? Maybe number of genome assemblies? Or contigs?
5. Line 223. Genome sequences of isolates?
6. Line 243. Antimicrobial Resistance by Class. It should be clearer. By class of drugs or classes of ARG genes?
7. In the discussion part, I suggest to discuss the frequency of abundance of these pathogens in the food and human clinical samples.
8. Also, there are nothing about localization of ARGs in these microbes. Usually, ARGs are mediated plasmids, MEGs and etc.
9. The current manuscript is called “Investigation of Antimicrobial Resistance in bacterial pathogens recovered from foods and humans – Analysis of 639,087 Bacterial Whole-Genome Sequences in the NCBI Pathogen Detection Database” and it is quite promising. But in the current form, the article looks like to more data analyses than experimental article. So, I suggest to authors, to change the name of article or to expand from the biological point.
